# High-Precision Combined Tidal Forecasting Model

**Jiao Liu, Guoyou Shi * and Kaige Zhu**

Key Laboratory of Navigation Safety Guarantee of Liaoning Province, Navigation College,
Dalian Maritime University, Dalian 116026, China; liujiaodmu@gmail.com (J.L.); zkg@dlmu.edu.cn (K.Z.)
* Correspondence: nsgi@dlmu.edu.cn; Tel.: +86-411-8472-5168

**Abstract:** To improve the overall accuracy of tidal forecasting and ameliorate the low accuracy of single harmonic analysis, this paper proposes a combined tidal forecasting model based on harmonic analysis and autoregressive integrated moving average–support vector regression (ARIMA-SVR). In tidal analysis, the resultant tide can be considered as a superposition of the astronomical tide level and the non-astronomical tidal level, which are affected by the tide-generating force and environmental factors, respectively. The tidal data are de-noised via wavelet analysis, and the astronomical tide level is subsequently calculated via harmonic analysis. The residual sequence generated via harmonic analysis is used as the sample dataset of the non-astronomical tidal level, and the tidal height of the system is calculated by the ARIMA-SVR model. Finally, the tidal values are predicted by linearly summing the calculated results of both systems. The simulation results were validated against the measured tidal data at the tidal station of Bay Waveland Yacht Club, USA. By considering the residual non-astronomical tide level effects (which are ignored in traditional harmonic analysis), the combined model improves the accuracy of tidal prediction. Moreover, the combined model is feasible and efficient.

**Keywords:** tidal level prediction; combined model; harmonic analysis method; Support Vector Regression (SVR); Autoregressive Integrated Moving Average Model (ARIMA)

## 1. Introduction

Tide is the periodic rising and falling of the sea level, and its fluctuations largely influence human lifestyle. Accurate real-time recording of tide level information is essential for ship navigation safety, the development and utilization of marine resources, and marine disaster mitigation and prevention [1]. Therefore, a simple and efficient tidal prediction method is urgently required. Based on their underlying prediction principles, tidal prediction methods are classified into traditional and intelligent prediction models.

Traditional tidal prediction models are mainly based on harmonic analysis [2,3]. Harmonic analysis for tide prediction was pioneered by Thomson in 1866, which was subsequently improved by Darwin, who formulated the equilibrium tide theory. Doodson determined the harmonic analysis constants by least-squares fitting the observed tidal data [4]. Yen smoothed the harmonic analysis constants by passing them through a Kalman filter [5]. After hundreds of years of development, harmonic analysis continues to be widely used in tidal prediction; however, this model only considers the astronomical tidal level affected by the tide-generating forces. Other environmental factors such as wind, pressure, and seabed topography, which exert nonlinear effects on the tidal level, are ignored. If the tide level is predicted by harmonic analysis alone, a large prediction error is incurred, which can be verified in Section 3.2.2 of this paper. Harmonic analysis also requires long-term historical data of tidal levels, which are generally precluded by the high cost of on-site monitoring equipment [6].

In today's artificial intelligence era, data prediction is performed by increasingly intelligent models. Owing to their strong adaptive learning ability and nonlinear-mapping ability, neural networks are

now widely used in tidal prediction. Many researchers have combined neural networks with related intelligent algorithms in their tidal prediction models, with much success.

The first prediction of diurnal and semidiurnal tides by an artificial neural network was attempted by Tsai et al. [7]; Lin et al. [8] proposed an adaptive neuro-fuzzy inference system for sea level prediction, which accounts for the tidal forces and thermal expansion of the ocean; and Jain et al. [9] developed a 24 h tidal prediction model based on a neural network, and applied it to the New Mangalore tidal station on the west coast of the Indian Ocean. However, neural networks generally require a large amount of training data, are easily trapped into local optima, lack universal applicability, etc. Support vector machines (SVM) have been widely used in prediction because they provide good nonlinear fitting with small input data volumes and are strongly generalizable [10]. Bhasin et al. [11] successfully forecasted the families and subfamilies of G-protein coupled receptors by an SVM-based approach; Xiong et al. [12] combined an SVM with a Hidden Markov model, and hence proposed a new framework of vehicle collision prediction; and Deris et al. [13] hybridized the SVM model with graded resolution, and predicted the surface roughness during abrasive water-jet machining. Oliveira et al. [14] proposed an evolutionary hybrid system composed of an exponential smoothing filter, the Autoregressive Integrated Moving Average Model (ARIMA), autoregressive (AR) linear models, and an SVR model, which has been proven to have good prospects in the forecasting field. Given this diversity of applications, the prospects of SVM in tidal prediction are high. Nevertheless, tidal prediction by SVM has rarely been reported.

To exploit these prospects, the current study proposes a tidal prediction model based on harmonic analysis and an autoregressive integrated moving-average–SVM for Regression (ARIMA-SVR): The model uses the typical time-series-processing model ARIMA and the SVR, with an excellent nonlinear-data regression performance, to predict the residual sequence generated by the prediction of harmonic analysis. The ARIMA-SVR prediction model is a data-driven model. It has unique advantages in solving numerical predictions, rebuilding highly non-linear functions, time series analysis, and so on. It does not need to consider the physical mechanism of the tidal formation process, but establishes a mathematical analysis of time analysis. By learning the given samples, we can find the statistical or causal relationship among the variables of water level, which has broad prospects in tidal prediction.

By using the complete data to extract information, including the astronomical tide level and the non-astronomical tide level, the proposed model greatly improves the accuracy of tidal prediction. The model was validated against the measured tide data at the port of Bay Waveland Yacht Club in Mississippi, USA. The verification proves that the combined model effectively ameliorates the low accuracy of a single model and provides effective tidal prediction.

## 2. Related Concepts

### 2.1. SVM and SVR

Pioneered by Vapnik in 1995, SVM is a nonlinear learning method with a solid theoretical foundation. Unlike other machine learning methods, such as neural networks, SVM implements the principle of Structural Risk Minimization (SRM) [15]. In convex quadratic programming problems, SVM seeks the best generalization performance by balancing the learning ability of the finite sample [16] against the complexity of the model. It has two variants: support vector classification (SVC) and support vector regression (SVR), which solve data classification and regression prediction problems, respectively. SVM architecture is shown in Figure 1, where $x(n)$ is the input independent eigenvalue and $K(X, X_i)$ $(i = 1, 2, \ldots, n)$ is the kernel function. The independent variable $x(n)$ is mapped to the high-dimensional feature space by the kernel function to realize linear regression in the feature space, and multiple linear regression is then performed in the high-dimensional feature space [17], and the output characteristic $Y$ is obtained.

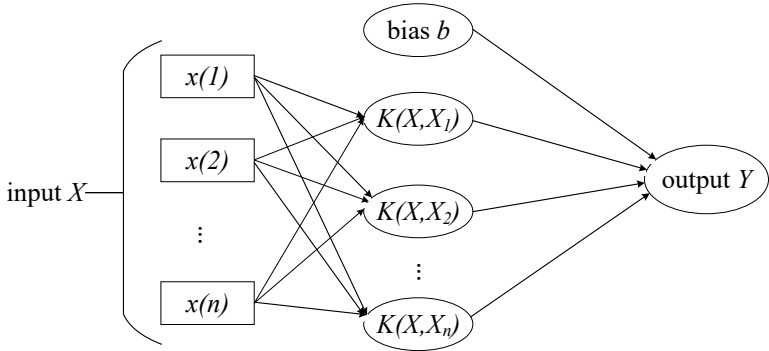

**Figure 1.** Architecture of a support vector machine (SVM).

In the present paper, the kernel is the popularly used radial basis function (RBF):

$$K(x, x_i) = \exp\left(\frac{-\|x - x_i\|^2}{2 \times \delta^2}\right) = \exp\left(-g\|x - x_i\|^2\right) \tag{1}$$

where $[x, x_i]$ is the initial low-dimensional feature space vector, $\delta$ is the inner bandwidth of the kernel function, $g$ is the parameter of the kernel function, and $\|\ \|$ is the 2-norm operator. By mapping the RBF kernel function, each sample $(x_i, y_i)$ is maximally fitted to the following linear SVR model:

$$Y_i = \langle \omega, x_i \rangle + b \ (\omega \in R^n, b \in R) \tag{2}$$

where $\langle \rangle$ is the inner product notation and incorporates the kernel function; the weight vector $\omega$ and bias $b$ are obtained by solving the following optimization problems:

$$\min_{\omega, b, \xi, \xi^*} \frac{1}{2}\|\omega\| + C\sum_{i=1}^{n}(\xi_i + \xi_i^*), \tag{3}$$

$$\text{s.t.} \begin{cases} y_i - (\omega x_i + b) \leq \varepsilon + \xi_i \\ (\omega x_i + b) - y_i \leq \varepsilon + \xi_i^* \\ \xi_i, \xi_i^* \geq 0 \end{cases} \tag{4}$$

The factor $C$ is the penalty factor, which compromises between the generalization performance and the training error, and $\varepsilon$ is the maximum tolerance beyond which the optimization fails. $\xi_i$ and $\xi_i^*$ are relaxation variables that avoid over-fitting during data training, ensuring a certain fault tolerance of the model.

### 2.2. PSO Algorithm

The particle swarm optimization (PSO) algorithm [18] is a swarm intelligence optimization algorithm proposed by Kennedy and Eberhart in 1995. Like the fish swarm algorithm and the ant colony algorithm, PSO optimizes the solution via group intelligence generated by the mutual cooperation of particles and information sharing. PSO also has a memory function for dynamically tracking the current search situation and adjusting the search strategy in real time [19]. Because of its simplicity, high efficiency, and lack of many parameter adjustments, PSO has been widely applied in fuzzy system control, parameter optimization of machine learning algorithms, and function optimization.

The PSO algorithm is fully described below.

Step 1: Initialize a group of particles in $D$-dimensional space. Each particle represents a set of potential optimal solutions to the optimization problem and is characterized by its position, velocity, and fitness. The position and velocity vectors are represented as $X_i = (X_{i1}, X_{i2}, X_{i3}, \ldots, X_{iD})$ and $V_i = (V_{i1}, V_{i2}, V_{i3}, \ldots, V_{iD})$, respectively.

Step 2: As the particles move in space, update their individual positions by tracking the individual extremum $P_{\text{best}}$ and the group extremum $G_{\text{best}}$. $P_{\text{best}}$ and $G_{\text{best}}$ represent the local and global fitness of the optimal particle positions, respectively.

Step 3: Calculate the fitness after updating the particle position. The $P_{\text{best}}$ and $G_{\text{best}}$ positions are updated by comparing the fitness of the new particle with those of the individual extremum and group extremum computed in Step 2. In each iteration, the particle adjusts its velocity vector based on the inertia vector $P_i = (P_{i1}, P_{i2}, P_{i3}, \ldots, P_{iD})$, the optimal empirical vector $P_g = (P_{g1}, P_{g2}, P_{g3}, \ldots, P_{gD})$, and its own experience. It then adjusts its position vector. The specific update formula is

$$V_{id}^{k+1} = \omega V_{id}^k + c_1 r_1 (P_{id}^k - X_{id}^k) + c_2 r_2 (P_{gd}^k - X_{id}^k) \tag{5}$$

$$X_{id}^{k+1} = X_{id}^k + V_{id}^{k+1} \tag{6}$$

where $c_1$ and $c_2$ are the learning factors, and $r_1$ and $r_2$ are random numbers between 0 and 1 (with $i = 1, 2, \ldots, n; d = 1, 2, \ldots, D$).

### 2.3. The Harmonic Analysis Method

Tidal forces can be regarded as the sum of forces during different periods; in other words, as the sum of many simple harmonic oscillations. The harmonic analysis method separates the harmonic constants (including amplitudes and phase lags) of each tidal component from the continuous observation data of tide heights [6]. Tidal height is calculated by summing the $m$ tidal components as follows:

$$h(t) = A_0 + \sum_{i=1}^{m} R_i \cos(\omega_i t - \theta_i) = A_0 + \sum_{i=1}^{m} (\alpha_i \cos \omega_i t + \beta_i \sin \omega_i t) \tag{7}$$

where $\alpha_i = R_i \cos \theta_i$; $\beta_i = R_i \sin \theta_i$; $A_0$ is the mean sea level; $R_i$ is the component amplitude; and $\omega_i$ and $\theta_i$ are the angular velocity and initial phase of the tidal components, respectively.

### 2.4. ARIMA

The ARIMA model is a typical time-series analysis and prediction model. From historical time-series data, we can build a dynamic model and predict the future trend of the data [20]. The ARIMA model is based on the autoregressive and moving average model, which is represented by the following formula:

$$Y_t = \mu + \sum_{i=1}^{p} r_i y_{t-i} + \varepsilon_t + \sum_{i=1}^{q} \theta_i \varepsilon_{t-i} \tag{8}$$

and it adds the following difference operator:

$$\begin{cases} \Delta Y_{t-i} = Y_{t-i} - Y_{t-i-1} = Y_{t-i} - LY_{t-i} = (1 - L)Y_{t-i} \\ \Delta^2 Y_{t-i} = \Delta Y_{t-i} - \Delta Y_{t-i-1} = (1-L)Y_{t-i} - (1-L)Y_{t-i-1} = (1-L)^2 Y_{t-i} \\ \Delta^d Y_{t-i} = (1-L)^d Y_{t-i} \end{cases} \tag{9}$$

In these expressions, $Y_t$ is the predicted value at time $t$; $\varepsilon_{t-i}$ and $\varepsilon_t$ denote the errors at times $t - i$ and t, respectively; and $Y_{t-i}$ is the measured value at time $t - i$. $\mu$ is a constant; $r_i$ is the autocorrelation coefficient; $\theta_i$ is the is the moving average coefficient, which is different from the initial phase of the tidal components $\theta_i$ mentioned in Formula (7); $d$ is the differential term; and $\Delta^d y_{t-i}$ is the time series after adding the $d$-order difference. $p$ is the order of the autoregressive model, which represents the lagged rank of the time series. $q$ is the order of the moving average model, which represents the lagged rank of prediction errors. The model must determine three parameters $(p, d, q)$. If the sequence is unstable, it should be transformed into a stable sequence by the $d$-order difference (Equation (9)), and $y_{t-i}$ in Equation (8) is replaced with $\Delta^d y_{t-i}$. If the original sequence is a stationary, $d$ is set to 0.

## 3. Combined Tidal Forecasting Model Based on Harmonic Analysis and ARIMA-SVR

*3.1. Prediction Steps*

As mentioned above, tides can be considered as the superposition of astronomical tidal levels and non-astronomical tidal levels. The astronomical tidal level is strongly periodic, being governed by the tide-generating force, whereas the non-astronomical variations in water level largely depend on climate, hydrology, wind, and other environmental factors, and exhibit a strong randomness [21]. Based on the above analysis, this paper establishes a combined tidal prediction model based on harmonic analysis and ARIMA-SVR. The prediction steps are described below.

The first step is to preprocesses the sample data. While acquiring tidal data, many factors may cause the data to become inaccurate and incomplete, resulting in noise interference. The preprocessing step removes the noise from the tidal sequence, thus restoring the tidal motions and improving the prediction accuracy. The preprocessing is performed by wavelet analysis theory.

Second, the height of the astronomical tide level is calculated via harmonic analysis. The harmonic constants are calculated by analyzing the historical observations of tidal heights, and the tidal heights are subsequently calculated. Meanwhile, the residual series generated by this method are collected as the sample data of the non-astronomical tidal level component.

Third, the nonlinear variations in water level are predicted by the ARIMA-SVR model. As the astronomical characteristics of the data have been processed in the astronomical tidal level component, the non-astronomical tidal levels change is reflected in the prediction residue. This prediction step is divided into several sub-steps: Exploiting the strong processing ability of ARIMA for time series, a single-step ARIMA model of non-astronomical tidal level sequences is established, which determines the input in the SVR forecasting model according to the lagged rank of the time series $p$ in ARIMA. The analysis of the time sequence in the ARIMA model confirms that the residual values from $t - 1$ to $t - p$ moment have a noticeable relevance with the values of moment $t$, which can be chosen as the input of the SVR forecasting model. That is, the residual values from $t - 1$ to $t - p$ moment are used to predict the value at time $t$. The second sub-step establishes a non-astronomical tidal level prediction model based on SVR, which has a strong nonlinear processing ability. As mentioned above, the input variable of SVR mode is

$$\text{Input} = \left\{ Y_{t-1}, Y_{t-2}, Y_{t-3}, \ldots, Y_{t-p'} \right\} \tag{10}$$

where $Y_{t-p'}$ represents the value at time $t - p'$. Additionally, the value of $p'$ is determined by the value of the lagged rank of the time series $p$ in the ARIMA model. Without considering other factors ($p'$ value is different only, other factors are the same), the prediction model error is the smallest when $p' = p$, which is verified in Section 3.2.4. Furthermore, the output variable is

$$\text{Output} = \{Y_t\} \tag{11}$$

The residual sequence generated by the harmonic analysis is normalized to avoid computational saturation. The data are normalized to the interval (0, 1) as follows:

$$Y_t = \frac{x_t - x_{\min}}{x_{\max} - x_{\min}} \ (x \in R^n, \ y \in (0,1)) \tag{12}$$

where $x_{\min} = \min(x)$ and $x_{\max} = \max(x)$. The optimal kernel function type, penalty factor $C$, and kernel function parameter $\delta$ are found by the PSO algorithm and then input to the SVR model for training. In the third sub-step, the test set samples are predicted by the trained model, and the tidal calculation values of the non-astronomical part are obtained.

In the fourth main step, the final tidal height is predicted via the equal-weight summation of the astronomical tidal level and the non-astronomical tidal level. The whole prediction process schematic is shown in Figure 2.

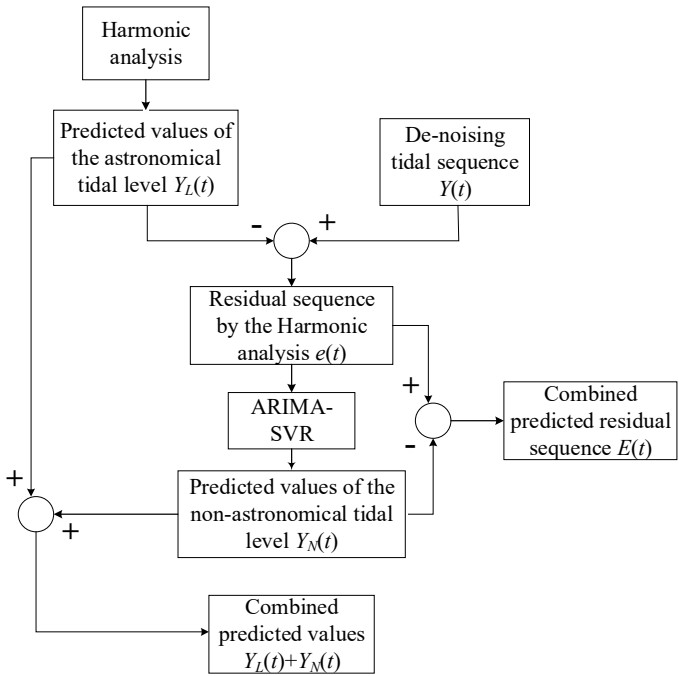

**Figure 2.** Schematic of the combined tidal prediction model.

The error in the prediction model was measured by the mean absolute error ($E_{\mathrm{MA}}$), the mean squared error ($E_{\mathrm{MS}}$), the root mean squared error ($E_{\mathrm{RMS}}$), and the correlation coefficient $r$. These four performance measures are respectively calculated as follows:

$$E_{\mathrm{MA}} = \frac{1}{n}\sum_{i=1}^{n}|(y-Y)|, \tag{13}$$

$$E_{\mathrm{MS}} = \frac{1}{n}\sum_{i=1}^{n}(y-Y)^2, \tag{14}$$

$$E_{\mathrm{RMS}} = \sqrt{\sum_{i=1}^{n}(y-Y)^2/n}, \tag{15}$$

$$r = \frac{\mathrm{Cov}(y,Y)}{\sqrt{\mathrm{Var}[y]\mathrm{Var}[Y]}}. \tag{16}$$

Here, $y$ and $Y$ are the measured and predicted tidal heights, respectively; $n$ is the number of tidal samples; $\mathrm{Cov}(y,Y)$ is the covariance between $y$ and $Y$; and $\mathrm{Var}[y]$ and $Var[Y]$ denote the variances in $y$ and $Y$, respectively.

### 3.2. Model Checking

The combined tide-forecasting model presented in this paper was checked against the observed tide level data at the Bay Waveland Yacht Club port in the USA, obtained from the website of the National Oceanic and Atmospheric Administration. The harmonic constants of four tidal constituents of the Bay Waveland Yacht Club tidal station are shown in Table 1. The tidal coefficient of the port is 10.767, indicating a diurnal tide-only one high and low tide each day. The tidal level of this port was predicted by the proposed model. The astronomical part of the tidal height was calculated via harmonic analysis, obtaining 720 tidal calculations at 1 h intervals over 30 consecutive days in November 2018. Meanwhile, the non-astronomical part of the water level variation was determined by the ARIMA-SVR model, wherein the training set was compiled from 744 prediction residual data of the astronomical tidal levels from GMT0000 on 1 October 1 2018 to GMT2300 on 31 October 2018, and 720 prediction

residual data from GMT0000 on 1 November 2018 to GMT 2300 on 30 November 2018, which were used as the test set for verifying the prediction results. Finally, both parts of the tidal calculation results were superimposed to obtain the predicted tidal levels throughout November 2018.

**Table 1.** The harmonic constants of four tidal constituents of the Bay Waveland Yacht Club tidal station.

| Constituent | Amplitude/m | Phase/° |
|:-----------:|:-----------:|:-------:|
| $M_2$ | 0.03 | 31.7 |
| $S_2$ | 0.026 | 36 |
| $K_1$ | 0.169 | 328.2 |
| $O_1$ | 0.154 | 325.1 |

### 3.2.1. Sample Data Preprocessing

The original waveform was constructed from the real-time signal sets (the tidal series measured from GMT0000 on 1 December 2017 to GMT2300 on 31 December 2017). The original waveform is shown in Figure 3. The waveform construction was based on the Sym8 wavelet basis function and two-layer wavelet decomposition. The threshold was selected using the heursure function, and de-noising was performed with a soft threshold function. The original and de-noised waveforms are compared in Figure 4, and the noise signal is shown in Figure 5. De-noising comparatively flattened the processed waveforms without changing their overall trend (Figure 4). The effectiveness of the de-noising pretreatment was verified in a comparative test (Section 3.2.4). The results show that de-noising improves the accuracy of the sample data.

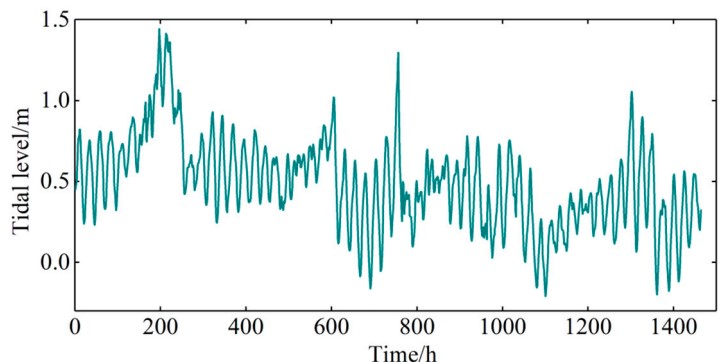

**Figure 3.** Sequence of measured tidal levels.

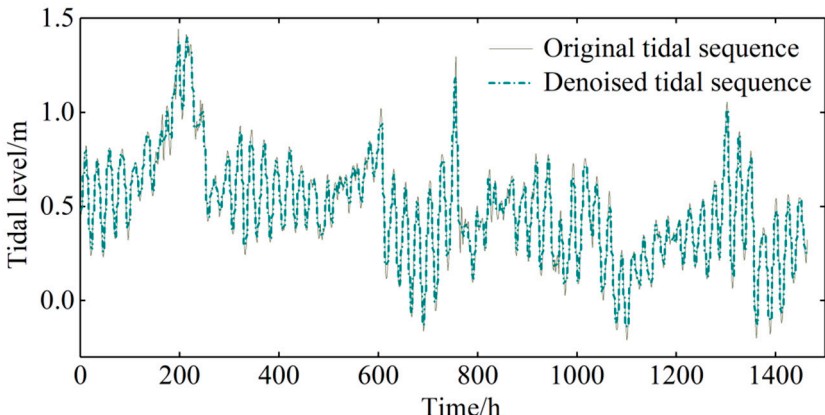

**Figure 4.** Comparison of the original and de-noised tidal level sequences.

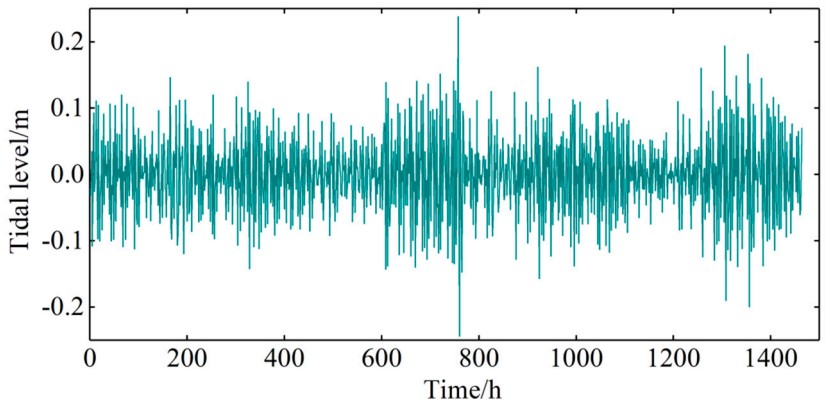

**Figure 5.** Sequence of the noise signal.

### 3.2.2. Analysis of Prediction Results of Astronomical Tide Level

The astronomical tidal level calculation used 11 main tidal constituents, namely, $M_2S$, $S_2$, $N_2$,$K_2$,$K_1$, $O_1$, $P_1$, $Q_1$, $M_4$, $MS_4$, and $M_6$. Figure 6 compares the astronomical tidal level predicted by the harmonic analysis method with the original (not de-noised) measured tidal level. Figure 7 is an error distribution chart of the tidal levels predicted by the harmonic analysis method alone, and Figure 8 plots the linear regression between the observations and predicted results of the harmonic analysis method. As indicated by the deviation of the best-fit line (red line in Figure 8) from the $Y = X$ line, the observed and predicted values were highly discrepant. The simple harmonic analysis introduced obvious errors at some points. The $E_{\text{RMS}}$ of the harmonic analysis method was determined as 0.180571 m. This large error is attributed to the oversimplified analysis method: the simple harmonic method only considers the influence of the tide-generating force and ignores the non-astronomical components of the changing water levels.

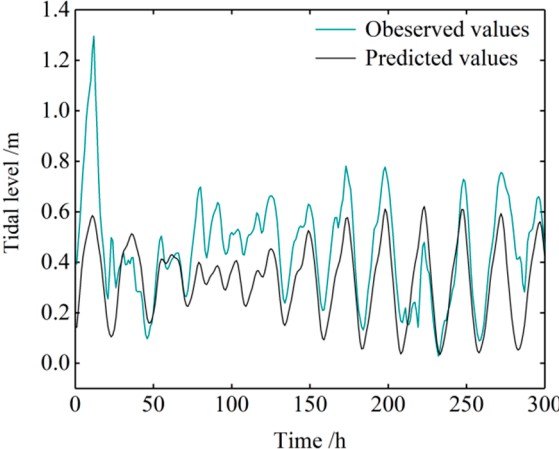

**Figure 6.** Comparison of tidal levels predicted by the harmonic analysis method alone and the observed data.

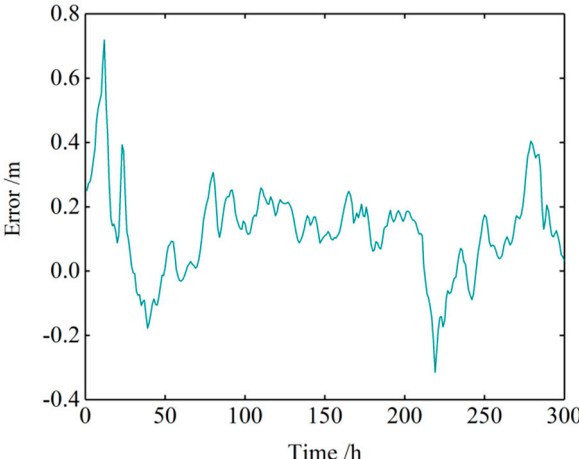

**Figure 7.** Error distribution map of the harmonic analysis prediction.

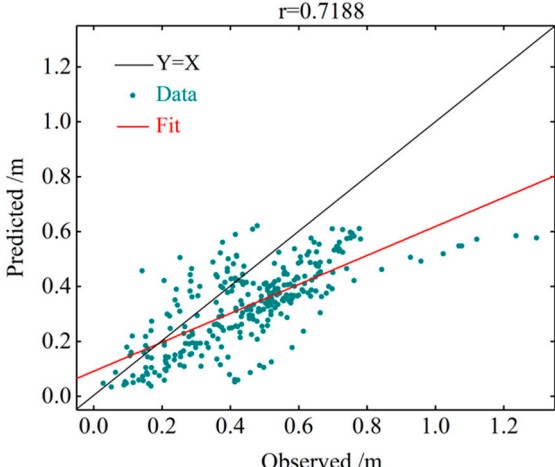

**Figure 8.** Linear regression of the harmonic analysis predictions versus the observed results. The red line is the best-fit line of the data. Along the black line, the predicted and observed values are equal, and the error is zero.

### 3.2.3. Analysis of Prediction Results of Non-Astronomical Tide Level

The model parameters of the non-astronomical part were determined by the single-step ARIMA model. To determine the difference term $d$, the stationarity of the sequence must be determined by visual processing. To this end, the 744 nonlinear data throughout October 2018 were processed by the augmented Dickey–Fuller (ADF) stationarity test. The parameters are shown in Table 2. Here, the p-value is the probability of significance test in statistics, which is different from the order of the autoregressive model $p$ mentioned in Formula (8). The $t$-statistics were below the critical values at the 1%, 5%, and 10% significance levels, and the p-values were close to 0, confirming that the tidal level sequence was a stationary sequence with $d = 0$ [22]. To determine the factors $p$ in the autoregressive term and $q$ in the moving average term, the sequence was evaluated by an autocorrelation function and a partial autocorrelation function [23], respectively. The evaluation results are shown in Figures 9 and 10, respectively. In the partial autocorrelation function plot (Figure 9), the sequence is mainly located in the confidence interval after the third order; i.e., it begins to truncate after the third order, and the autoregressive term $p$ is thus 3. After establishing $p$, the independent variables of the sample set in the SVR model are set to the tide levels at times $t - i$ ($i = 1, 2, 3$), and the dependent variable is the tide level at time $t$.

**Table 2.** Parameters determined in the ADF unit root test.

| Parameter Type | t-Statistic | p-Value | 1% Level | 5% Level | 10% Level |
|---|---|---|---|---|---|
| Parameter values | −3.679002 | 0.0002 | −2.568196 | −1.941266 | −1.616402 |

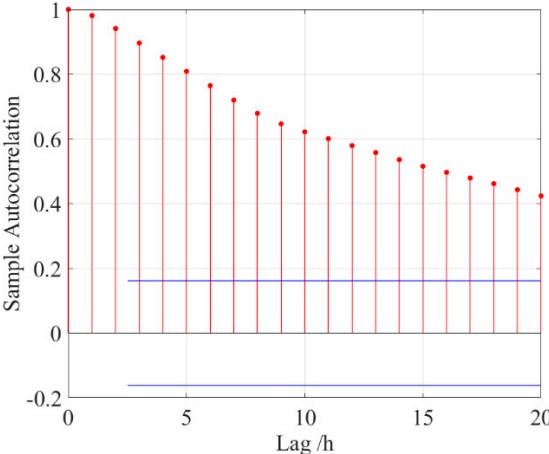

**Figure 9.** Autocorrelation analysis of the residual tidal level data. The area between the two blue lines is the confidence interval.

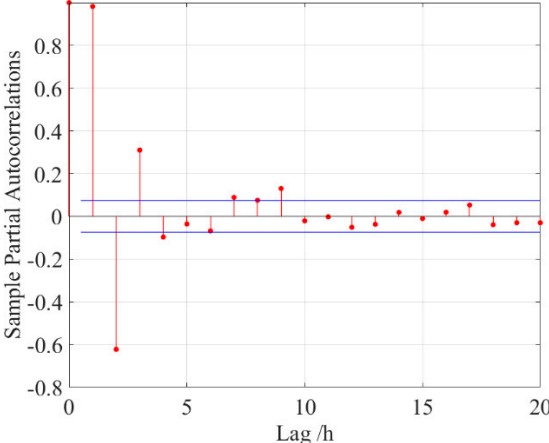

**Figure 10.** Partial autocorrelation analysis of the residual tidal level data. The area between the two blue lines and the coordinate axis is the confidence interval.

As is evident in the autocorrelation plot (Figure 8), the tidal level time series is tailing, and the moving average term $q$ is 0. In summary, the sequence establishes the ARIMA $(3, 0, 0)$ model. The prediction process for the non-astronomical tidal part is shown in Figure 11.

To reduce the influence of the order of magnitude of the sample on the prediction accuracy, the data in the training and test sets were normalized to the interval (0,1), and the RBF was selected as the kernel function. The parameters (penalty factor $C$ and kernel function parameter $g$) were optimized by the PSO algorithm. The group termination algebra was set to 200; the population number was set to 20; and the learning factors $C_1$ and $C_2$ were set to 1.5 and 17, respectively. The optimization results are shown in Table 3. The searched optimal parameters were input to the SVM model as the training data. After training, the test data were input to the model, and the predicted results were compared with the real results. The comparisons and their relative errors are displayed in Figures 12 and 13, respectively.

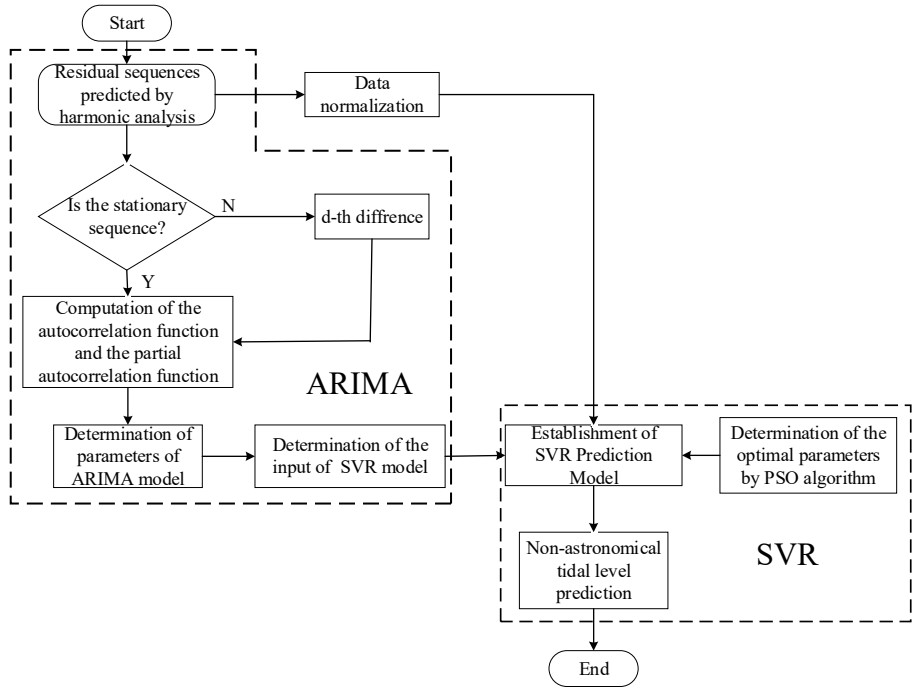

**Figure 11.** Schematic of the non-astronomical tidal level prediction part.

**Table 3.** Optimal parameters detected by PSO.

| Parameter | Penalty Factor $C$ | Kernel Function Parameter $g$ |
|---|---|---|
| SVR model | 20.4906 | 798.0125 |

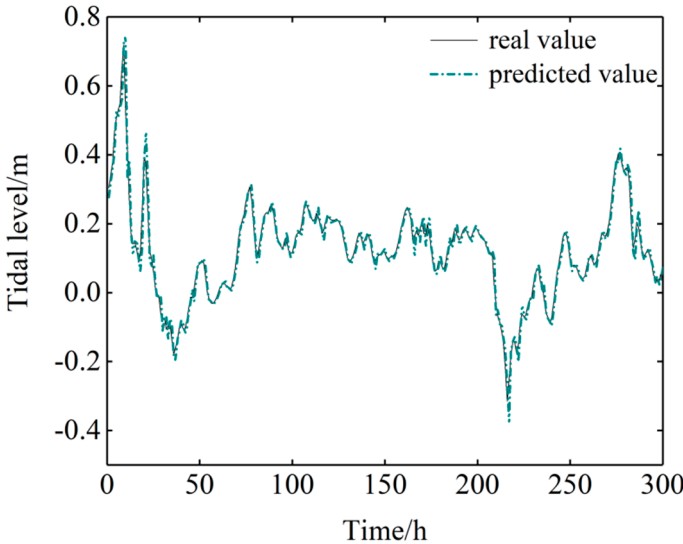

**Figure 12.** Comparison of the observations and predicted non-astronomical tidal level computed by the SVR model.

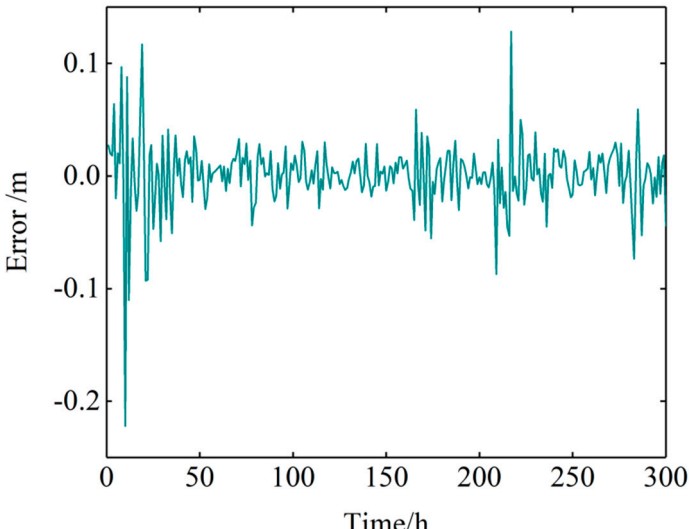

**Figure 13.** Error distribution map of the non-astronomical tidal level.

### 3.2.4. Analysis of Prediction Results of the Combined Model

Next, the astronomical tidal level and the non-astronomical tidal level were linearly added to obtain the overall tidal prediction throughout November 2018. Figure 14 compares the predicted tidal levels with the observed (not de-noised) data, and Figure 15 plots the predicted errors. The combined model yielded much more accurate results than the pure harmonic analysis method. The $E_{RMS}$ of the combined model was 0.022293 m, which is obviously smaller than that in the harmonic analysis method. Figure 16 linearly regresses the predictions of the combined model against the observed (not de-noised) data. The combined model clearly predicted the observed tidal level with a high accuracy.

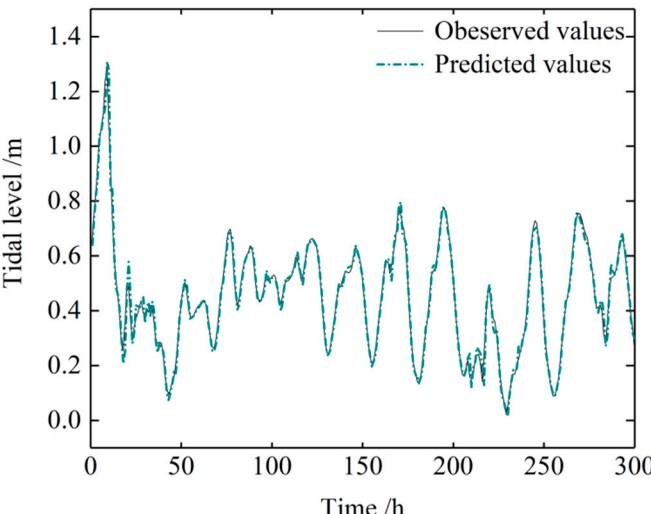

**Figure 14.** Comparison of the (not de-noised) observations and the water levels predicted by the combined model.

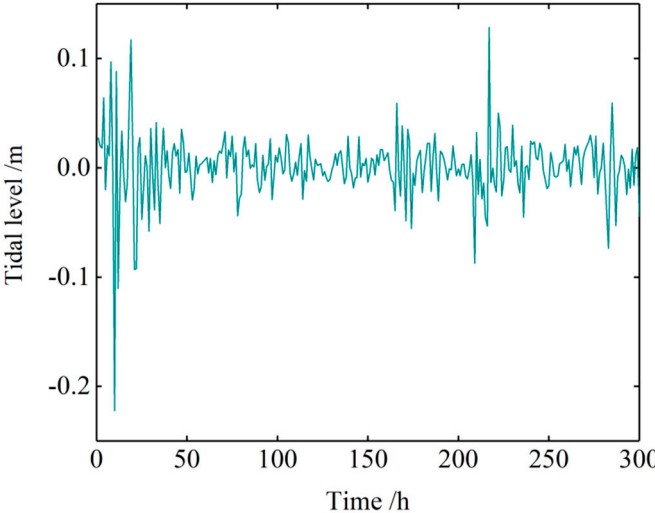

**Figure 15.** Error distribution map of the combined model.

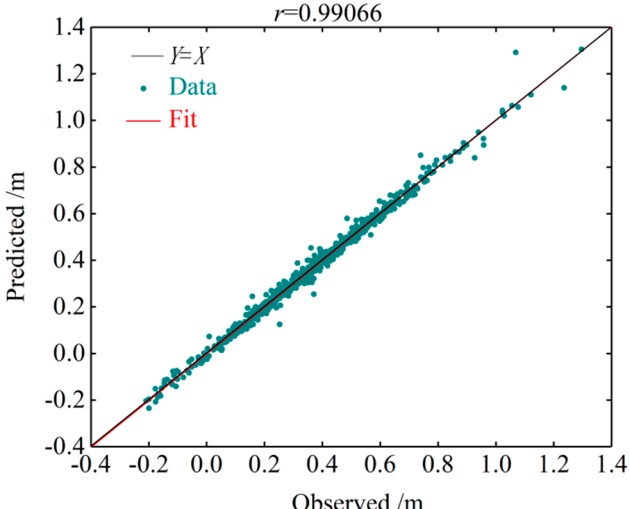

**Figure 16.** Linear regression plot of the predicted versus observed tidal heights. The predicted results were calculated by the combined model.

To verify the effect of de-noising, the tide level was predicted using the original data (without de-noising) and the predicted error was calculated by the above steps. The errors in the predictions are compared with those of the de-noised data in Table 4. Clearly, the wavelet transform smoothed the data and improved the prediction accuracy. This result confirms the feasibility and effectiveness of de-noising the sample data prior to analysis.

**Table 4.** Comparison of the predicted tide level errors with and without de-noising.

| Data | $E_{MA}/m^2$ | $E_{MS}/m$ | $E_{RMS}/m$ |
|---|---|---|---|
| With de-noising | 0.0152928 | 0.0005121 | 0.0226293 |
| Without de-noising | 0.0245718 | 0.0015876 | 0.0398451 |

To further verify the prediction accuracy of the combined model, the total tidal levels at the Bay Waveland Yacht Club station were predicted by single harmonic analysis, the combined model, the SVR model, and another common method called back propagation neural networks (the BP model) [24]. The parameters of the SVR model were optimized by the PSO algorithm. For a fair comparison, the sample data and parameters were identical in all methods. The prediction performances of the four

methods are compared in Table 5. The proposed combined model required a longer training time, but yielded more accurate tidal predictions with lower errors than the other models.

**Table 5.** Performance comparison of different models simulating the tidal behavior at Bay Waveland Yacht Club. The second column lists "$p'$", which refers to the fact that when the prediction time step is 1, the residual values from $t - 1$ to $t - p'$ moment are used to predict the value at time $t$. The last column lists the processing time required for the various methods.

| Model | $p'$ | $E_{MA}/m^2$ | $E_{MS}/m$ | $E_{RMS}/m$ | $r$ | Time/s |
|---|---|---|---|---|---|---|
| Harmonic Analysis | - | 0.1439167 | 0.0326057 | 0.1805706 | 0.52892 | 32.547131 |
| BP | 3 | 0.1381779 | 0.0194776 | 0.1395623 | 0.61248 | 7.404504 |
| PSO-SVR | 3 | 0.0415324 | 0.0023897 | 0.0498613 | 0.97381 | 56.04157 |
| Combined model | 3 | 0.0152928 | 0.0005121 | 0.0226293 | 0.99066 | 148.28581 |

The prediction accuracy of a model depends on the size of the training set. Accordingly, the prediction accuracies of the SVR and combined models were compared on training sets with different sample sizes (samples collected over 1, 3, 6, or 12 months). In this comparison, the test set remained fixed. ARIMA modeling is performed on the training sets of different sample sizes, and the lagged order of time series $p$ is determined, and the input of the model is determined thereby. Again, the data were the tidal levels at the Bay Waveland Yacht Club tidal station. The prediction results of the SVR model and the combined model are shown in Tables 6 and 7, respectively. As the sample size increased, the $E_{RMS}$ of the tidal levels predicted by the SVR model (Table 6) changed substantially around 0.049, whereas those of the combined model (Table 7) fluctuated around 0.022 m. By contrast, the error indicators of tidal prediction were lower in the combined model than in the SVR model. The combined model thus exhibited a more accurate and stable prediction performance than the SVR model alone, within a significantly lower runtime than the SVR model. This result confirms the efficiency of the combined model.

**Table 6.** Sample-size comparison tidal prediction errors in the SVR model. The second column lists "$p'$", which refers to the fact that when the prediction time step is 1, the total tidal level from $t - 1$ to $t - p'$ moment is used to predict the total tidal level at time $t$.

| The Size of Sample Set | $p'$ | $E_{MA}/m^2$ | $E_{MS}/m$ | $E_{RMS}/m$ | $r$ | Time /s |
|---|---|---|---|---|---|---|
| One month | 3 | 0.0415324 | 0.0023897 | 0.0498613 | 0.9738146 | 56.042 |
| Three months | 2 | 0.0413146 | 0.0022134 | 0.0495362 | 0.9738124 | 37.312 |
| Six months | 4 | 0.0412874 | 0.0021045 | 0.0494135 | 0.9738113 | 64.617 |
| Twelve months | 3 | 0.0411763 | 0.0020983 | 0.0493245 | 0.9738137 | 143.982 |

**Table 7.** Sample-size comparison tidal prediction errors in the combined model. The second column lists "$p'$", which refers to the fact that when the prediction time step is 1, the residual water level from $t - 1$ to $t - p'$ moment is used to predict the non-astronomical tidal level at time $t$.

| The size of Sample Set | $p'$ | $E_{MA}/m^2$ | $E_{MS}/m$ | $E_{RMS}/m$ | $r$ | Time /s |
|---|---|---|---|---|---|---|
| One month | 3 | 0.0152928 | 0.0005121 | 0.0226293 | 0.9906614 | 148.286 |
| Three months | 2 | 0.0152994 | 0.0005184 | 0.0227172 | 0.9906592 | 174.217 |
| Six months | 4 | 0.0152613 | 0.0005127 | 0.0226981 | 0.9906452 | 209.576 |
| Twelve months | 2 | 0.0151472 | 0.0005094 | 0.0226242 | 0.9906601 | 251.880 |

As mentioned above, the Bay Waveland Yacht Club tidal station experiences a diurnal tide. To test the combined model on different tidal types and stations, the harmonic analysis, SVR, and combined models were trained on the tidal level data from four stations with different tidal types, and their predictive performances were evaluated in each case. Nawiliwili, The Battery, and Texas Point tidal stations were selected for tidal prediction comparison experiments, which have different tidal types.

The ARIMA model was established for the tidal data of different tidal stations to determine the lagged rank of the time series $p$, so the input of the non-astronomical tidal part was determined. The error results are shown in Table 8. At all four stations, the combined model outperformed the pure harmonic analysis and pure SVR models. In order to further measure the prediction accuracy, the relative magnitude of the astronomical tide and non-astronomical tide parts of the sample data of four tidal station was calculated, and the following Table 8 was obtained. As shown in Tables 8 and 9, it can be seen that the larger the relative magnitude of the astronomical tide, the higher the accuracy of the harmonic analysis. The harmonic analysis method is suitable for predicting tidal stations with a high relative magnitude of astronomical tides.

**Table 8.** Tidal-type comparisons of prediction errors in the harmonic analysis, SVR, and combined models. The third column lists "$p'$", which refers to the fact that when the prediction time step is 1, the value from $t-1$ to $t-p'$ moment is used to predict the value at time $t$.

| Tidal Stations | Tidal Type | $p'$ | Combined Model | | SVR Model | | Harmonic Analysis | |
|---|---|---|---|---|---|---|---|---|
| | | | $E_{RMS}$/m | $r$ | $E_{RMS}$/m | $r$ | $E_{RMS}$/m | $r$ |
| Nawiliwili | Semidiurnal mixed tide | 3 | 0.0157421 | 0.9967487 | 0.0637458 | 0.9267340 | 0.0517735 | 0.9760012 |
| The Battery | Semidiurnal tide | 2 | 0.0345787 | 0.9945872 | 0.2547214 | 0.8998741 | 0.2661074 | 0.9014375 |
| Texas Point | Diurnal mixed tide | 2 | 0.0424531 | 0.9857817 | 0.0845512 | 0.9459475 | 0.1908219 | 0.7382920 |
| Bay Waveland Yacht Club | Diurnal tide | 3 | 0.0152928 | 0.9906615 | 0.0498613 | 0.9738146 | 0.1805706 | 0.7188010 |

**Table 9.** The relative magnitude of the astronomical tide and the un-astronomical tide parts of four tidal stations.

| Tidal Stations | The Astronomical Tide | The Non-Astronomical Tide |
|---|---|---|
| Nawiliwili | 88.1535% | 11.8465% |
| The Battery | 85.9369% | 14.0631% |
| Texas Point | 74.2308% | 25.7692% |
| Bay Waveland Yacht Club | 60.406% | 39.594% |

The input of the SVR also significantly affects the prediction accuracy of the combined models. To verify the robustness of the input of SVR in the combined model determined according to the lagged rank $p$ of the residual sequence in the ARIMA model, the combined model was trained on the tide level data from the Bay Waveland Yacht Club tidal station, and its predictive performance was compared for different inputs. The prediction results are shown in Table 10. Here, the first column lists "$p'$", which refers to the fact that when the prediction time step is 1, the residual values from $t-1$ to $t-p'$ moment is used to predict the value at time $t$. Increasing the $p'$ from 1 to 3 reduced the error in the combined model, but increasing the $p'$ further increased the error. The minimized error at $p'=3$ is consistent with the test results of the lagged rank $p$ of the residual sequence in the ARIMA model. However, even at the largest $p'$ ($p'=12$), the error was lower in the combined model than in the harmonic analysis method. Therefore, the combined model is more accurate and more suitable for short-term tidal level prediction than the simple harmonic model.

**Table 10.** The input of SVR model comparisons of prediction accuracy of the combined method at Bay Waveland Yacht Club Tidal Station. The first column lists "$p'$", which refers to the fact that when the prediction time step is 1, the residual values from $t-1$ to $t-p'$ moment is used to predict the value at time $t$.

| $p'$ | $E_{MA}$/m$^2$ | $E_{MS}$/m | $E_{RMS}$/m | $r$ | Time /s |
|---|---|---|---|---|---|
| 1 | 0.0218577 | 0.0009071 | 0.0301173 | 0.9916444 | 29.233 |
| 2 | 0.0163588 | 0.0005606 | 0.0236780 | 0.9948650 | 40.472 |
| 3 | 0.0152412 | 0.0005076 | 0.0225293 | 0.9953641 | 130.600 |
| 4 | 0.0153038 | 0.0005076 | 0.0225293 | 0.9953641 | 169.639 |
| 6 | 0.0154409 | 0.0005186 | 0.0227718 | 0.9952334 | 281.034 |
| 12 | 0.0793346 | 0.0073769 | 0.0858892 | 0.9904756 | 556.413 |

## 4. Conclusions

This study analyzes predictions of tide level by common analysis techniques and those of a proposed method that combines simple harmonic analysis with autoregressive moving-average–support vector regression (ARIMA-SVR). To better capture the tidal dynamic properties and improve the prediction accuracy, the original tidal wave data were de-noised by wavelet threshold analysis. The method fully utilizes the strong time-series processing ability of ARIMA and the outstanding nonlinear regression predictions of the SVR. Harmonic analysis and ARIMA-SVR in the combined model predict the astronomical tide level and the non-astronomical tidal level, respectively. In the non-astronomical component of the data, the data were normalized and the ARIMA-SVR parameters were optimized by the PSO algorithm, which improved the overall prediction accuracy of this component. In experimental tests, the combined model achieved a higher prediction accuracy than any of its composite models used on their own, as well as the BP neural network. Therefore, the proposed model ameliorates the low precision associated with single prediction models. Accordingly, it promises broad applications in future tidal prediction. However, as mentioned in the introduction, harmonic analysis as a part of the combined model has the problem of the high cost of obtaining large amounts of data from on-site monitoring equipment. This problem has not been solved in the model proposed in this paper and is the next problem that needs to be solved.

**Author Contributions:** Software, Writing-Original Draft Preparation, Writing-Review & Editing, J.L.; Supervision, G.S.; Validation, K.Z.

**Funding:** This research was funded by the National Natural Science Foundation of China, grant number 51579025; the Natural Science Foundation of Liaoning Province, grant number 20170540090; and the Fundamental Research Funds for the Central Universities, grant number 3132018306.

**Conflicts of Interest:** The authors declare no conflict of interest.

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
