# Peer review of "High-Precision Combined Tidal Forecasting Model"

_algorithms, doi:10.3390/a12030065_

Round 1
Reviewer 1 Report
The article is written very logically. Initially, theoretical considerations concerning the analyzed phenomenon were presented, followed by the achievements of science in this subject. The next section presents the models used.The algorithm proposed in the work has been described in a legible and comprehensible way.
Ending, the article is very interesting, and the mathematical methods used and their modification is at a high scientific level.
Author Response
Thank you very much for your careful reading of our manuscript.
Thank you very much for your confirmation,.
I will continue to work hard!
Reviewer 2 Report
see attached

Author Response
I would like to express our great appreciation to reviewers for their positive and constructive comments and suggestions on our manuscript. I have studied comments carefully and have made revision which was highlighted in the updated manuscript. Those comments are all valuable and very helpful for revising and improving our paper, as well as the important guiding significance to our researches. The responses to each of the points raised by the reviewers are in red after each of their comments. Please check the file "Response to Reviewer 2 Comments" for detail!

Round 2
Reviewer 2 Report
The revised manuscript has, for the most part, adequately addressed my comments from the first review round.
Some points arise from the revised version:
In response to my query about the “Time” parameter in what was Table 4 in the original manuscript, the authors revised the caption of what is now table 5 to read “The second column lists “i” refer to the “i” of the moment of t-i (i=1, 2, 3, …) in the input of SVR model”. This is somewhat confusing, as the text (e.g. line 202) makes it clear that, for BP, PSO-SVR and the combined model, “i” takes on a range of values up to p, so ALL values of i=1,…,p are used. In which case, surely “p” is the appropriate label rather than “i”. Otherwise the wording suggests that only values at t-3 were used as input.
Then in Tables 6 and 7, results are reported from applying SVR to different lengths of record, while in Table 8 results from 4 different sites are compared. In these cases, it is stated that only values at t-1 were used as input, effectively choosing to use p=1 rather than the value p=3 derived from applying ARIMA (at least to the 30-day Bay Waveland record). This choice is not explained, and should be, as it deviates from the method outlined in Section 3.1 and Section 3.2.3.
In light of this choice, if the authors wish to reserve “p” as the order of autoregressive model in the ARIMA, perhaps a slightly different symbol (p’?) could be used (e.g. in Section 3.1, 3.23, Table 5 caption) as the possibly-different value chosen for the SVR model.
Author Response
Thank you for your comments concerning our manuscript.The responses to each of the points raised by you are in red after each of their comments. Please check the file "Report Notes" for detail!

Round 3
Reviewer 2 Report
The changes made in the latest revision have clarified the issues that I raised previously.
As a further minor amendment for consistency with those changes, I would suggest that at line 515 “Increasing the i from 1 to 3 reduced the error in the combined model, but increasing the i further …” should change to “Increasing p’ from 1 to 3 reduced the error in the combined model, but increasing p’ further …”
Also, the Acknowledgements section just contains instructions from the style template. If the authors have no Acknowledgements to make, this section should be omitted (this was a point I overlooked in previous versions).
Author Response
Thank you for your comments concerning our manuscript. I accept your suggestion. The responses to each of the points raised by you are in red after each of their comments. Please check the file "Report Notes" for detail!
